# A High Performance and Low Latency Deep Spiking Neural Networks Conversion Framework

## Abstract

Spiking Neural Networks (SNN) are promised to be energy-efficient and achieve Artificial Neural Networks (ANN) comparable performance through conversion processes. However, a converted SNN relies on large timesteps to compensate for conversion errors, which as a result compromises its efficiency in practice. In this paper, we propose a novel framework to convert an ANN to its SNN counterpart losslessly with minimal timesteps. By studying the errors introduced by the whole conversion process, an overlooked inference error is reveald besides the coding error occured during converting. Inspired by the quantization aware traning, a QReLU activation is introduced during training to eliminate the coding error theoretically. Furthermore, a buffered non-leaky-integrate-and-fire neuron that utilizes the same basic operations as in conventional neurons is designed to reduce the inference error. Experiments on classification and detection tasks show that our proposed method attains ANNs level performance using only 16 timesteps. To the best of our knowledge, it is the first time converted SNNs with low latency demonstrate their capability to achieve high performance on nontrivial vision tasks. Source code will be released later.

## 1 Introduction

Recently, Spiking Neural Networks (SNN) have attracted a great deal of researchers' attention due to their essential advantages, such as spatio-temporal information processing capability and energy-efficient with low power consumption. Unlike conventional Artificial Neural Networks (ANN) communicating with every neuron between layers during every inference step, SNN only passes sparse events when particular neurons are fired. This compute-on-demand property of SNN is very friendly to low-power hardware, and suitable for neuromorphic platforms, such as event-based camera [11] and brain-inspired chips [32]. Such dedicated bionic systems can significantly reduce memory usage and energy consumption, which is very appealing to a wide range of applications like autonomous mobile robots.

Despite the promising characteristics, it is still a challenging problem to train high-performance deep SNNs for practical tasks. Directly applying the backpropagation approach is infeasible because of the non-differentiable nature of the spike activity. Taking advantage of the ANN's success, conversion methods have been proven to be a very promising direction in achieving high performance SNN. Cao et al. [4] first convert ANNs to SNNs by demonstrating the relationship between spiking neuron and ReLU function for rate-based methods. Diehl et al. [9] propose a weight normalization approach to mapping weights from ANN to SNN. Later, Rueckauer et al. [39] give a theoretical analysis on ANN-SNN conversion. A reset-by-subtraction IF neuron is proposed to better handle the accuracy degradation during the conversion. They also propose the weight normalization algorithm that uses percentile function instead of max value to achieve better performance. Kim et al. [23] propose a channel-wise normalization to further reduce the performance gap between ANNs and SNNs .

Although great progress has been made, those converted SNNs suffer from *accuracy-latency trade-off* [9, 31]. As a result, the converted SNNs need longer time to achieve comparable precision with their counterpart ANN during inference and compromises the efficiency in practice. To deal with this problem, hybrid methods [37, 36] are proposed to further finetune a converted SNN with approximated gradients to decrease the inference time. On the other hand, for the theoretical training gaps between ANNs and SNNs, Ding et al. [10] propose Rate Norm to better choose the scale factor and introduce extra loss to decrease the latency. Deng and Gu [8] utilize training samples to transfers the weights to the target SNN by combining threshold balance and soft-reset mechanisms. However, these works only focus on reducing the errors introduced by mapping feature values to spike trains.

With an in-depth analysis of the entire coversion process, we recognize two types of conversion errors, namely coding error and inference error, that degrade the performance of SNN. As other conversion approaches mainly focus on the coding errors, our motivation is to reduce both the coding and inference errors. Inspired by the analysis in Rueckauer et al. [39], we notice that the SNN with step function resembles quantized ANNs. By establishing a strict equivalency between spiking neuron and quantized activation function, the coding error can be eliminated completely under limited timesteps. Further, we observe that the assumption in rate-based conversioin is inconsistent with the actual neuron behavior and the SNN's dynamics during the inference. Most of the previous conversion-related literature overlook this problem, which limits the performance of the algorithms and leads to longer simulation time to achieve comparable results of the original ANN.

In this paper, we propose a novel High Performance Conversion (HPC) method to drastically close the gap between ANNs and SNNs. Specifically, a QReLU activation is proposed to eliminate the coding error. Since QReLU is applied to the ANN training phase, no extra effort is required during the conversion. Besides, a noval buffered non-leaky IF neuron is designed to compensate for the inference error and retain SNN's efficiency during inference. Utilizing a new form of activation function and novel efficient spiking neuron, SNN obtained by our proposed approach achieves high performance and low latency simultaneously. With only 16 timesteps, HPC achieves state-of-the-art performance across classification and detection tasks. Our major contributions are summarized as:

- An overlooked inference error is discovered. The conversion errors including both coding errors and inference errors are theorically analized to clearify the reason of performance degration during conversion.

- A novel high performance conversion method is proposed to deal with the two types of conversion errors. The coding errors are eliminated completely in theory and the inference error problem is relieved to a great extent.

- Comprehensive experiments on different tasks demonstrate the efficacy of the proposed method. To the best of our knowledge, it is the first time converted SNN achieves ANN level performance while maintaining high efficiency and low latency across various tasks.

## 2    Related Work

Up to now, there exist two main categories of supervised training strategies for SNN: approximated error backpropagation methods, and conversion methods.

Our proposed work is mainly related to ANN-SNN conversion methods with rate-based coding [9, 35, 4, 39]. The goal of conversion approaches is to remain the high performance of ANN in SNN framework. Recently, Xing et al. [44] extend the territory of conversions by adding supports to more structures such as softmax activation and residual block. Instead of using longer timesteps to aproximate ANN accurately, conversion errors are analized and compensated by adjusting the parameters of converted SNN [27]. Rate Norm is proposed [10] to better choose the scale factor. Deng and Gu [8] utilize training samples to transfers the weights to the target SNN by combining threshold balance and soft-reset mechanisms. As a general approach, Kim et al. [23] show that conversion can be applied to detection tasks. However, only coding error are considerd in those work. We notice that the inference error or "unevenness error" is described in the parallel work [3]. However, the "unevenness error" is ignored during the analysis of conversion error in their work.

Other researchers [2, 29, 21] try to approximate the gradient to train the SNN by replacing the threshold function with other functions. A comprehensive summary of surrogated gradient backpropagation methods can be found in [30]. Huh [20] introduce differentiable synapse and neuron models to

SNNs. Since spike trains are normaly sparse, gradients in SNNs are naturely sparse compared to traditional ANNs. An sparse spiking gradient descent approach is introduced [34] to speed up the backpropgation. To enable very deep spiking neural network, modified residual module with element-wise functions [14] is proposed to mimic directly gradient path of ResNet [17]. Rather than compute the approximated gradient, a neuron with extra inner state is introduced in [42] to estimate the gradient. Different from [42], the buffered neruon proposed in this literature utlizes extra inner state to compensate for inference error. Backpropagation trained SNNs show the potential of achieving ANN level performance. However, extra efforts are required to train the SNN properly, making it hard to train and limit its application on real world tasks.

Recently, combined methods [37, 36] emerges to alleviate the drawbacks that conversion methods require larger timesteps to achieve competative performance. These methods often contain two steps: conversion and fine-tuning. The SNN converted in the first step is served as a weight initialization for a further fine-tuning procedure using backpropagation. With the help of roughly mapped weights, several epochs of fine-tuning yield nearly loss-less performance with fewer timesteps [37]. However, compared to conversion methods, extra efforts are required to fine-tune the SNN. Also, the backpropagation imposes additional limitations to the hybrid approach, which further restrains its application.

## 3   Methods

In this section, we first analyse the theoretical equivalency and gaps between a target SNN and an original ANN. Then, a quantized activation is proposed to minimize the coding error during the conversion. A buffered non-leaky IF neuron is presented to reduce the inference error. Finally, we integrate the proposed conversion framework to further decrease the difference between SNN and ANN.

### 3.1   Theoretical Errors and Analysis

Following Rueckauer et al. [39], we here illustrate the equivalency between SNN and ANN and introduce two types of errors when using the conversion methods with rate-based coding. The target SNN shares a similar overall architecture with the original ANN except two significant discrepancies. First, neurons in the SNN communicate with each other by spike trains $S(t)$ where $S(t) \in \{0, 1\}$ for each timestep $t \in \{1, ..., T\}$, while the activation functions in ANN utilize real-value. $T$ is the total timesteps. Second, a non-leaky integrate-and-fire (IF) neuron is used instead of the ReLU activation in conventional ANN. The membrane function of the non-leaky IF neuron $j$ can be described as:

$$V_j(t) = V_j(t-1) + V_{th} \sum_{i \in \mathcal{N}_j} w_{ij}^s S_i(t) - V_{th} S_j(t). \tag{1}$$

where $V_j(t)$ is the membrane potential of the $j$th neuron and $t$ is the timestep. $\mathcal{N}_j$ is the set of all input neurons in the last layer connecting to $j$th neuron. $w_{ij}^s$ is the weight and connection strength between the input $i$th neuron and current $j$th neuron. $V_{th}$ is the threshold for the neuron to fire a signal. $S_i(t)$ and $S_j(t)$ are input and the resulting output spike trains of current neuron, respectively. The firing signals are generated as follow:

$$S_j(t) = \begin{cases} 1, & \text{if } V_j'(t) \geq V_{th} \\ 0, & \text{otherwise.} \end{cases} \tag{2}$$

where $V_j'(t) = V_j(t-1) + V_{th} \sum^{\mathcal{N}_j} w_{ij}^s S_i(t)$ denotes the membrane potential before signal firing and potential reset operation.

Considering ANN-SNN conversion and rate-based coding, we can compute spike rate as $r = \sum_{t=1}^{T} S(t)/T$ for a total timestep $T$. The goal of a conversion process is to derive a function transfers the parameters of an ANN to the SNN as $w^s = f(w^a)$ so that each neuron in the SNN produces a spike rate $r \propto a$ after $T$ timesteps, where $a$ is the corresponding activation value of original networks.

To this end, we can derive the spike count of the output spike train by summing Eqn. 1 over total timesteps $T$:

$$\sum_{t=1}^{T} S_j(t) = \sum_{i \in \mathcal{N}_j} w_{ij}^s \sum_{t=1}^{T} S_i(t) - \frac{V_j(T) - V_j(1)}{V_{th}}, \tag{3}$$

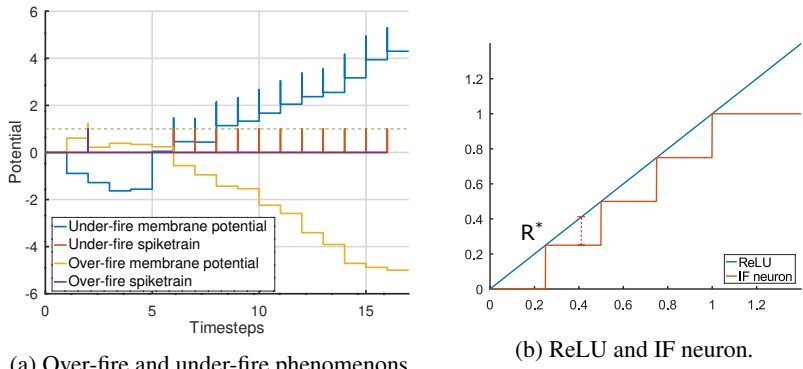

(a) Over-fire and under-fire phenomenons

(b) ReLU and IF neuron.

Figure 1: For each step of the membrane potential line in the figure, the left horizontal line indicates $V(t-1)$. The peak point indicates $V'(t)$, and the right horizontal line indicates $V(t)$ which is transferred to the next step. For over-fire spike train(purple), $\sum_{t=1}^{T} S_j(t) = 1$ while $\sum_{t=1}^{T} S_j^*(t) = 0$. For under-fire spike train(orange), $\sum_{t=1}^{T} S_j(t) = 11$ while $\sum_{t=1}^{T} S_j^*(t) = 15$.

where $V_j(1)$ is the initial membrane potential. For convenience, we set $V_j(1) = 0$ in this work without loss of generality. Eqn. 3 gives the relation between the total spike count of current neuron and its input neurons' spike count besides the internal membrane potential.

For an ideal conversion, the spiking neuron intergrates membrane potential uniformly accross time and introduces no error during inference. Under this ideal condition, the output spike count should fullfill the following equation,

$$\sum_{t=1}^{T} S_j^*(t) = \left\lfloor \max\left(\frac{V_a}{V_{th}}, 0\right) \right\rfloor, \tag{4}$$

where $V_a = V_{th} \sum_{i \in \mathcal{N}_j} w_{ij}^s \sum_{t=1}^{T} S_i(t)$ denotes the accumulated membrane potential. Please refer to the appendix for detail proof. Assuming Eqn. 4 hold, we can derive the ideal output spike rate by averaging Eqn. 4 over timesteps $T$:

$$r_j^* = \max\left(\sum_{i \in \mathcal{N}_j} w_{ij}^s r_i, 0\right) - R^*. \tag{5}$$

where $R^*$ is defined as follow.

$$R^* = \begin{cases} \frac{\frac{V_a}{V_{th}} - \left\lfloor \frac{V_a}{V_{th}} \right\rfloor}{T}, & \text{if } V_a > 0 \\ 0, & \text{otherwise.} \end{cases} \tag{6}$$

For the ideal ANN-SNN conversion situation, when $R^*$ is approaching zero, the formal expression of a simple IF neuron becomes similar to the ReLU function in processing information. Thus, the weight of a SNN can be transformed from its counterpart ANN directly. Since spike rate $r^*$ is bounded, a linear mapping function is utilized to convert ANN weights $W^s = f(W^a) = \frac{\lambda_i}{\lambda_j} W^a$, in which $\lambda_i$ is the scale factor for layer $i$.

### 3.1.1 Coding Error

As shown in Eqn. 5, $R^*$ can be viewed as a *coding error* term for the ideal conversion process during the ANN-SNN weight mapping. As $\frac{V_a}{V_{th}} - \left\lfloor \frac{V_a}{V_{th}} \right\rfloor < 1$ for $V_a > 0$, the upper bound of the conversion error $R^* < 1/T$ continuously decreases when the timesteps is set increasingly. Due to errors accumulated between layers, the converted SNNs with deep layers require large timesteps $T$ to maintain high fidelity and performance [39, 23]. On the other hand, the large timesteps impair the efficiency of the SNN. This phenomenon is also known as accuracy-latency trade-off [9, 31].

### 3.1.2 Inference Error

Besides the conversion error, the assumption in Eqn. 4 can be contravened easily due to essential nature of SNN neurons during the inference. Intuitively, the inference process of SNN is quite dynamic and spiking neurons are unable to accumulate membraine potential uniformly accross timesteps as Eqn. 4, which leads to the divergence of overall output spike counts. We refer to this type of error as *inference error*.

We conclude two situations where the Eqn. 4 is violated during inference: over-fire and under-fire phenomenons. Over-fire indicates the situation where number of total generated spikes is greater than number of expected spikes $\sum_{t=1}^{T} S_j(t) > \sum_{t=1}^{T} S_j^*(t)$. Volatile membrane potential may cross the threshold $V_j'(t) \geq V_{th}$ and cause spike $S_j(t) = 1$ that makes the number of generating spikes exceeding the number of expected spikes. Conversely, under-fire refers to the situation where insufficient spikes have been generated $\sum_{t=1}^{T} S_j(t) < \sum_{t=1}^{T} S_j^*(t)$. With inadequate spikes, under-fire neurons will have membrane potential large than $V_{th}$ at the end of inference. Fig. 1a shows an example of these two situations.

### 3.2 Quantized ReLU Activation

To deal with the coding error, we propose a Quantized ReLU activation function in ANN training stage to accurately transform the weights to SNN with limited timesteps. According to Eqn.4, we can reformulate Eqn. 5 in a different way :

$$r_j^* = \frac{\left\lfloor T \max(\sum^{\mathcal{N}_j} w_{ij}^s r_i, 0) \right\rfloor}{T}. \tag{7}$$

Let $x_s = \sum^{\mathcal{N}_j} w^s r_i$ be the weighted sum of input spike rates in SNNs. Taking maximal spike rate into consideration, IF neuron(Fig. 1b) can be described with step function as follow,

$$IF(x_s) = \begin{cases} 0, & \text{if } x_s \leq 0 \\ r^{max}, & \text{if } x_s \geq r^{max} \\ \frac{\lfloor T x_s \rfloor}{T} & \text{otherwise.} \end{cases} \tag{8}$$

The $r^{max}$ is the maximum spike rate of a spike train, which is 1 for conventional spike trains. The proposed function is strictly equivalent to the IF neuron instead of increasing total timesteps $T$ to approximate the ReLU function with a fine-grain step function. By mapping spike rate and activation value $a = \lambda r$, the proposed activation function is derived through Eqn. 8 :

$$QReLU(x_a, \lambda) = \lambda IF(\frac{x_a}{\lambda}) = \begin{cases} 0, & \text{if } x_a \leq 0 \\ \lambda, & \text{if } x_a \geq \lambda \\ \frac{\lfloor \frac{T}{\lambda} x_a \rfloor}{\frac{T}{\lambda}}, & \text{otherwise.} \end{cases} \tag{9}$$

We name the proposed activation function as QReLU, since it can be seen as a quantized version of the ReLU function. The approximate gradients are given by the PACT algorithm [6] originated from STE [1]:

$$\frac{\partial QReLU(x_a, \lambda)}{\partial \lambda} = \begin{cases} 0, & \text{if } x_a < \lambda \\ 1, & \text{otherwise.} \end{cases} \tag{10}$$

$$\frac{\partial QReLU(x_a, \lambda)}{\partial x_a} = \begin{cases} 1, & \text{if } 0 < x_a < \lambda \\ 0, & \text{otherwise.} \end{cases} \tag{11}$$

The scale factor $\lambda$ here is also referred to as quantization boundary and is learned during the ANN training process. Guaranteed by the success of quantization techniques [1, 6], a neural network consisting of QReLU is capable of achieving similar performance compared to the original ANN. By

mapping weights from an ANN equiped with QReLU activation, the coding error $R^*$ is eliminated in theory.

### 3.3 Buffered Non-leaky IF Neuron

For the inference error, we observe that reduing over-fire problem alone can boost performance siginificantly based on our experiments. To further improve our conversion framework, we propose a novel buffered non-leaky IF neuron to relieve the over-fire situation to compensate for the inference errors.

The proposed buffered neuron utilizes a recurrent mechanism to regulate the spikes. In the proposed neuron, an additional buffered potential $V^b$ is introduced as follows:

$$V_j^b(t) = V_j^b(t-1) + V_{th} S_j(t).$$  (12)

Similar to the membrane potential, the initial buffer potential $V_j^b(1)$ is set to zero for convenience. The spike generating process is described as

$$S_j(t) = \begin{cases} 1, & \text{if } V_j'(t) > V_{th} \\ -1, & \text{if } V_j'(t) < 0 \text{ and } V_j^b(t-1) \geq V_{th} \\ 0, & \text{otherwise.} \end{cases}$$  (13)

Intuitively, the proposed Buffered Non-leaky IF Neuron generates negative spikes to regulate the total spikes count for the inference errors. Thus, the over-fire problem can be alleviated. Please refer to appendix for detail proof. As same as ordinary IF neurons, only addition operation and threshold functions are involved in the proposed neuron. The inference efficiency is preserved since the extra power required by transmitting sign bit between neurons is negligible [45].

### 3.4 Integration with Multiple Bits Spike Train

To further improve the performance of the proposed conversion framework, we integrate shift-based multi-bits spike train into the proposed framework. Unlike the previous work [45] which compressed an existing SNN to speed up the inference, we extend the definition of the spike as $S(t) \in \{2^n | n \in \mathbb{N}, 2^n \leq S^{max}\}$ to facilitate the shift operation directly during the conversion. The multi-bits spike train version of Eqn.13 is as follow:

$$S_j(t) = \begin{cases} A\left(\left\lfloor \frac{V_j'(t)}{V_{th}} \right\rfloor\right), & \text{if } V_j'(t) > V_{th} \\ A\left(\min\left(\left\lfloor \frac{V_j'(t)}{-V_{th}} \right\rfloor, \left\lfloor \frac{V_j^b(t-1)}{V_{th}} \right\rfloor\right)\right), & \begin{array}{l} \text{if } V_j'(t) < 0 \\ \text{and} \\ V_j^b(t) \geq V_{th} \end{array} \\ 0, & \text{otherwise,} \end{cases}$$  (14)

where the adjust function $A(\cdot)$ is defined as

$$A(x) = 2^{\lfloor \log_2(\min(x, S^{max})) \rfloor}.$$  (15)

Multiple bits spike train carries more information than ubiquitous on-off spikes [5, 22, 33, 47]. However, expensive multiplications are required during the integration with non-uniform spikes. To avoid multiplication, we employ the $log_2$ adjust function and shift operation [45] as an alternative and achieve high energy efficiency. The adjust function can be implemented with a conditional function.

For QReLU, as the maximal ratio coding is equal to maximal spike counts, $r^{max} = \sum_{t=1}^{T} S^{max}/T = S^{max}$, the multi-bits version of Eqn. 9 can be described as follow:

Table 1: Classification performance compared to other methods in CIFAR10 and ImageNet-1k

| Model | Arch | SNN Top-1 | Δ | T |
|---|---|---|---|---|
| CIFAR10 | | | | |
| Rueckauer et al. [39] | 4 Conv, 2 Linear | 90.85% | -1.06% | 400 |
| Sengupta et al. [40] | ResNet20 | 87.46% | -1.64% | 2500 |
| Lee et al. [26] | VGG9 | 90.45% | - | 100 |
| Kim and Panda [24] | VGG9 | 90.50% | - | 25 |
| Wu et al. [43] | 5 Conv, 2 Linear | 90.53% | - | 12 |
| Han et al. [16] | ResNet20 | 91.36% | -0.11% | 2048 |
| Rathi et al. [37] | ResNet20 | 92.22% | -0.93% | 250 |
| Rathi and Roy [36] | ResNet20 | 92.14% | -0.65% | 25 |
| Zheng et al. [48] | ResNet19 | 93.16% | - | 6 |
| Yu et al. [47] | 6 Conv, 2 Linear | 93.90% | -0.23% | 300 |
| Yan et al. [46] | VGG19 | 92.48% | -0.08% | 600 |
| Deng and Gu [8] | ResNet20 | 92.41% | 0.09% | 16 |
| Ding et al. [10] | PreActResNet18 | 91.96% | -1.10% | 64 |
| Bu et al. [3] | ResNet18 | 94.82% | -1.22% | 8 |
| Bu et al. [3] | ResNet18 | 95.92% | -0.12% | 16 |
| **Ours** | **ResNet18** | **95.13%** | **-0.17%** | **8** |
| **Ours** | **ResNet18** | **95.14%** | **-0.04%** | **16** |

| Model | Arch | SNN Top-1 | Δ | T |
|---|---|---|---|---|
| ImageNet | | | | |
| Rueckauer et al. [39] | VGG16 | 49.61% | -14.28% | 400 |
| Sengupta et al. [40] | VGG16 | 69.96% | -0.56% | 2500 |
| Han et al. [16] | VGG16 | 73.09% | -0.40% | 4096 |
| Rathi et al. [37] | VGG16 | 65.19% | -4.16% | 250 |
| Rathi and Roy [36] | VGG16 | 66.52% | -3.56% | 25 |
| Deng and Gu [8] | VGG16 | 55.80% | -16.6% | 16 |
| Bu et al. [3] | VGG16 | 50.97% | -23.32% | 16 |
| Bu et al. [3] | VGG16 | 68.47% | -5.82% | 32 |
| **Ours** | **VGG16** | **73.65%** | **-1.69%** | **8** |
| **Ours** | **VGG16** | **75.96%** | **-0.07%** | **16** |
| Sengupta et al. [40] | ResNet34 | 65.47% | -5.22% | 2000 |
| Han et al. [16] | ResNet34 | 65.47% | % | 4096 |
| Fang et al. [14] | ResNet34 | 67.04% | - | 4 |
| Zheng et al. [48] | ResNet34 | 63.72% | - | 6 |
| Li et al. [27] | ResNet34 | 64.51% | -11.12% | 32 |
| Li et al. [27] | ResNet34 | 74.61% | -1.05% | 256 |
| Bu et al. [3] | ResNet34 | 59.35% | -14.97% | 16 |
| Bu et al. [3] | ResNet34 | 69.37% | -4.95% | 32 |
| **Ours** | **ResNet34** | **73.20%** | **-0.52%** | **8** |
| **Ours** | **ResNet34** | **74.46%** | **0.08%** | **16** |

Table 2: Detection performance

| Model | Arch | SNN AP50 | Δ | T |
|---|---|---|---|---|
| PASCAL VOC | | | | |
| Kim et al. [23] | tiny-yolo | 51.83% | -1.18% | 5000 |
| **Ours** | **tiny-yolo** | **65.20%** | **-0.13%** | **16** |
| **Ours** | **tiny-yolo** | **65.73%** | **-0.06%** | **32** |
| MS COCO | | | | |
| Kim et al. [23] | tiny-yolo | 25.66% | -0.58% | 5000 |
| **Ours** | **tiny-yolo** | **38.6%** | **-1.1%** | **16** |
| **Ours** | **tiny-yolo** | **39.2%** | **-0.6%** | **32** |

Table 3: SNN computation efficiency.

| Methods | Architecture (timesteps) | Operation ratio | SNN/ANN energy ratio |
|---|---|---|---|
| CIFAR10 | | | |
| Rathi and Roy [36] | ResNet20(25) | 2.55 | 0.51 |
| Ours | ResNet18(8) | 0.92 | 0.19 |
| Ours | ResNet18(16) | 1.69 | 0.34 |
| ImageNet | | | |
| Rathi and Roy [36] | VGG16(25) | 1.62 | 0.32 |
| Ours | VGG16(8) | 1.92 | 0.39 |
| Ours | VGG16(16) | 3.03 | 0.62 |

$$QReLU(x_a, \lambda^b) = \begin{cases} 0, & \text{if } x_a \leq 0 \\ \lambda^b, & \text{if } x_a \geq \lambda^b \\ \dfrac{\lfloor \frac{P}{\lambda^b} x_a \rfloor}{\frac{P}{\lambda^b}}, & \text{otherwise.} \end{cases} \tag{16}$$

Here the $\lambda^b = \lambda S^{max}$ is the quantization boundary and $P = TS^{max}$ is the *representation power*. As implied by the definition of the representation power, even the minimal shift extension with $S^{max} = 2$ can maintain the same representation power with half timesteps. This phenomenon is named *strength-latency trade-off*.

## 4 Experiments

In this section, we conduct experiments on both classification and detection tasks to demonstrate the efficiency and effectiveness of the proposed method. Since the equivalency of features and spike rates are universal throughout the entire network, real-valued input is directly utilized in the converted SNN. We select a unity threshold $V_{th} = 1$ for all our experiments. Max spike value $S^{max} = 2$ is used for all experiments if not stated otherwise. Other implementation details can be found in the appendix.

### 4.1 Visual Object Recognition and Detection Performance

We test our proposed High Performance Conversion framework (HPC) on visual object recognition and detection tasks. A ResNet18 [17] like architecture is utilized as backbone networks to conduct experiments on the CIFAR10 [25] dataset and a VGG16 [41] like network is trained on the ImageNet [7] dataset. Table 1 reports the classification accuracy of the proposed HPC compared

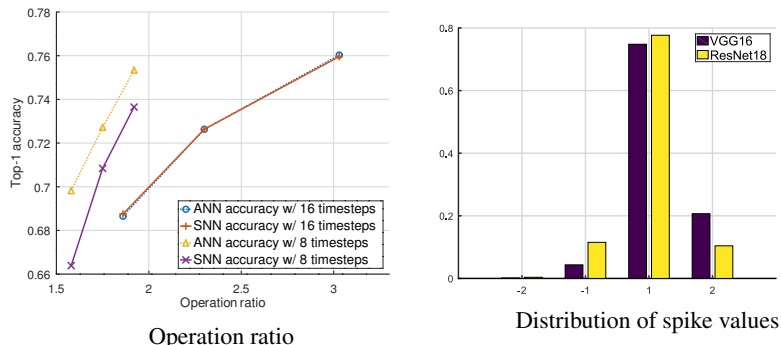

Operation ratio               Distribution of spike values.

Figure 2: Left: Operation ratio and performance with ImageNet dataset under 16 timesteps. Data points with same operation ratio are converted SNN and its corresbonding ANN. Right: Distribution of spike values.

with other methods. Top-1 accuracy is used to evaluate the performance. Since the performance of converted SNN is closely related to the ANN baseline, a performance drop $\Delta$ is also reported. As illustrated in Table 1, the proposed HPC can perform nearly lossless conversion while maintaining low inference latency.

As a general method, we also test our HPC with tiny-yolo [38] structure on both PASCAL VOC [13] and MS COCO [28] datasets. Table 2 summarizes the AP50 performance of the proposed method. We can see that our proposed HPC can achieve state-of-the-art conversion performance within 32 timesteps, which is over 100 times faster than the previous SNN detection work [23].

## 4.2 Energy Efficiency

To evaluate the energy efficiency of the proposed method, *operation ratio* is adopted to measure the overall efficiency as described in Eqn. 17. In contrast to actual energy consumption which may vary between hardware devices [32], counting-based metric [36] is straightforward and representative for different methods. For a fair comparison, integration operations triggered by spikes for both membrane potential and buffer potential are counted in our experiments.

$$operation\ ratio = \frac{\#(\text{addition ops in SNN})}{\#(\text{flops in ANN})}. \tag{17}$$

Operation ratios, as well as estimated energy consumption ratio, are summarized in Table 3. Here we roughly estimate that multiplication consumes 4 times more energy than addition according to Horowitz [18]. The results on the CIFAR10 dataset show a strong correlation between timesteps and energy efficiency. To further study energy efficiency, we test the relationship between operation ratio and networks performance. As depicted in Fig. 2a, under the same timesteps setting, the spike ratio continuously decreases with the overall performance. Our experiments confirm a general relationship between inference energy and performance, which holds even under constant latency.

Fig. 2b depicts the distributions of spike values in classification experiments. Without loss of generality, we conclude that only $20\%$ of the spikes require shift operation. As measured by Gudovskiy and Rigazio [15], the shift operation only cost $10\%$ of energy consumed by addition operation. Thus, multi-bits spike only use $2\%$ extra energy, which is negligible.

## 4.3 Ablation Study

To reveal the effectiveness of the proposed framework, we conduct ablation studies on both classification and detection tasks. Specifically, we remove the quantized training technique and the buffered IF neuron sequentially and compare the results. The baseline ANN is trained and fixed for experiments without QReLU component, meanwhile quantized ANN networks are trained separately for different timestep configurations. Without the QReLU, the scale factor is calculated using the percentile function with the parameter empirically set to 99.0. Channel-wise normalization is utilized to ensure competitive performances. Since conventional converted SNN requires large timesteps to

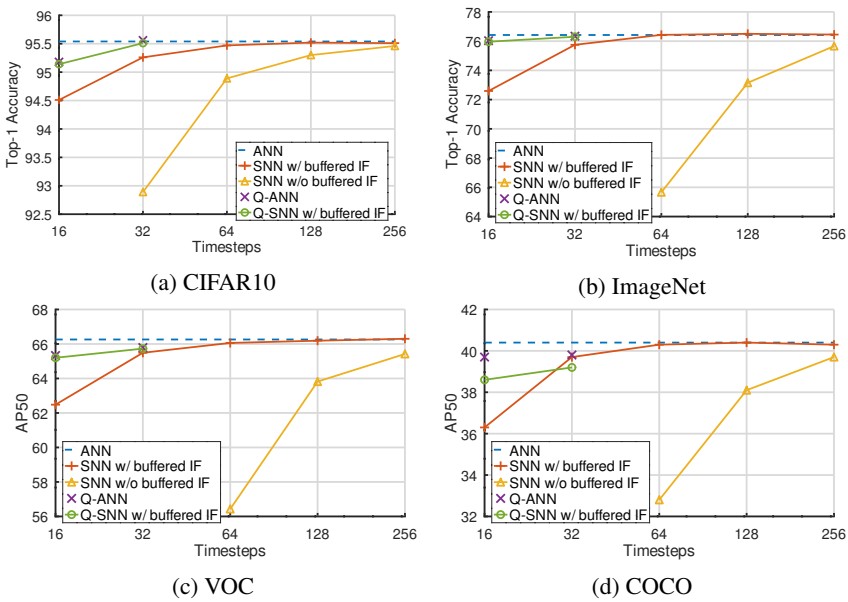

Figure 3: Ablation study of proposed techniques. Q-ANN denotes ANNs trained with QReLU, Q-SNN denotes SNNs converted from ANN trained with QReLU. SNN w/ buffered IF means that buffered non-leaky integrated and fire neuron are used. SNN w/o buffered IF are conventional SNNs with out the proposed neuron.

achieve plausible performance, we only report those performances with 32 or more timesteps for the CIFAR10 dataset and 64 for others.

Fig. 3 validates the effectiveness of the proposed techniques. Compared to conventional SNNs, the SNN integrates with buffered neurons only requires a quarter of timesteps to reach the same performance. As the buffered neuron mitigates only the over-fire problem, we conclude that the over-fire situation is the primary cause of performance degradation during inference. The introduction of quantized training further enhances the performance of converted SNN under limited timesteps. Similar trends can be observed across experiments.

## 5 Conclusion and Limitation

In this paper, we proposed a novel High Performance Conversion (HPC) method to simultaneously achieve high-performance SNN inference with low latency. Efforts had been made to reduce both coding errors and often overlooked inference errors. A novel QReLU activation was proposed to eliminates the conversion error. Meanwhile, a buffered non-leaky IF neuron was designed to mitigate the over-fire problem and boost the inference performance while maintaining its simplicity and efficiency. For the first time, efficient SNN converted without extra fine-tuning revealed its capability to achieve state-of-the-art performances in nontrivial vision tasks.

However, there are limitations. Since no actual hardware is involved, the efficiency is estimated using energy consumed by simple operations. As computing platform evolves rapidly, it is beyond our reach to accomplish real neuromorphic hardware which can be researched in further work.

For furture work, we believe further study on novel structures such as attention mechanisms SE [19] and transformer [12] can extend the current framework and may lead to better understanding of the underlying perception mechanism. Moreover, applications of SNN on video-like input that achieves ANN level performance with great efficiency are another promising direction.

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
