# Supplemental Materials: A High Performance and Low Latency Deep Spiking Neural Networks Conversion Framework

In this appendix, we first discuss the overall algorithm and the multi-bits version of our proposed method. Proofs are also provided to validate the ideal output spike count and the buffered neuron and over-fire situation. Then, we give more experiments for object detection with spike camera. Finally, we provide details about our experimental settings.

## A   Proofs and Implementation

### A.1   Overall Algorithm

The proposed conversion algorithm can be summarized in Algorithm 1. The buffered neuron is described in Algorithm 2.

The QReLU is given as follow:

$$QReLU(x_a, \lambda^b) = \begin{cases} 0, & \text{if } x_a \leq 0 \\ \lambda^b, & \text{if } x_a \geq \lambda^b \\ \frac{\left\lfloor \frac{P}{\lambda^b} x_a \right\rfloor}{\frac{P}{\lambda^b}}, & \text{otherwise.} \end{cases} \tag{1}$$

The multi-bits spike train is generated as follow:

$$S_j(t) = \begin{cases} A\left(\left\lfloor \frac{V'_j(t)}{V_{th}} \right\rfloor\right), & \text{if } V'_j(t) > V_{th} \\ A\left(\min\left(\left\lfloor \frac{V'_j(t)}{-V_{th}} \right\rfloor, \left\lfloor \frac{V^b_j(t-1)}{V_{th}} \right\rfloor\right)\right), & \begin{array}{l} \text{if } V'_j(t) < 0 \\ \text{and} \\ V^b_j(t) \geq V_{th} \end{array} \\ 0, & \text{otherwise,} \end{cases} \tag{2}$$

where the adjust function $A(\cdot)$ is defined as

$$A(x) = 2^{\lfloor \log_2(\min(x, S^{max})) \rfloor}. \tag{3}$$

### A.2   Proof of output spike count under ideal condition

The dynamics of IF neuron is described as follow:

$$V_j(t) = V_j(t-1) + V_{th} \sum_{i \in \mathcal{N}_j} w^s_{ij} S_i(t) - V_{th} S_j(t). \tag{4}$$

---

**Algorithm 1** Conversion algorithm

---

**Input**: Artificial neural network
**Parameter**: timesteps $T$, maximum spike rate $S^{max}$
**Output**: Spiking neural network

1: Train ANN with QReLU described in Eqn. 1.
2: Adjust the ANN parameters according to learned scale factors $W^s = \frac{\lambda_i}{\lambda_j} W^a$ where $\lambda = \frac{\lambda_b}{S^{max}}$.
3: Transfer parameters to spiking neural network with buffered non-leaky IF neurons.
4: **return** Spiking neural network

---

---

**Algorithm 2** Buffered neuron

---

**Input**: membrane potential $V_j(t-1)$, buffered potential $V_j^b(t-1)$, input spikes $S_i(t)$
**Parameter**: maximum spike rate $S^{max}$
**Output**: membrane potential $V_j(t)$, buffered potential $V_j^b(t)$ and output spikes $S_j(t)$

1: Integrate input to membrane potential $V_j'(t) = V_j(t-1) + V_{th} \sum^{\mathcal{N}_j} w_{ij}^s S_i(t)$.
2: Generate spikes according to Eqn. 2.
3: Reset membrane potential $V_j(t) = V_j'(t) - V_{th} S_j(t)$ and buffer potential as Eqn. 12.
4: **return** $V_j(t)$, $V_j^b(t)$ and $S_j(t)$

---

$$S_j(t) = \begin{cases} 1, & \text{if } V_j'(t) \geq V_{th} \\ 0, & \text{otherwise,} \end{cases} \tag{5}$$

14 where $V_j'(t) = V_j(t-1) + V_{th} \sum^{\mathcal{N}_j} w_{ij}^s S_i(t)$ denotes the membrane potential before signal firing
15 and potential reset operation.

16 Ideally, membrane potential is intergrated uniformnly as $V_{th} \sum_{i \in \mathcal{N}_j} w_{ij}^s S_i(t) = C$ across timesteps
17 $t \in \{1, ...T\}$. Combining Eqn. 4 and Eqn. 5, we can derive that

$$V_j(t) = \begin{cases} V_j'(t) - V_{th} \geq 0, & \text{if } V_j'(t) \geq V_{th} \\ V_j'(t) = V_j(t-1) + C, & \text{otherwise} \end{cases} \tag{6}$$

18 For $V_a > 0$, we can infer $C > 0$. For $V_j(t-1) \geq 0$, since $V_j(t-1) + C > 0$, we can have that
19 $V_j(t) \geq 0$ for $t \in \{1, ...T\}$. With $V_j(1)$ set to 0, we can conclude that $V_j(T) \geq 0$.

20 Divide Eqn. 4 by $V_{th}T$, we can rewrite the Eqn. 4 as:

$$r_j = \sum_{i \in \mathcal{N}_j} w_{ij}^s r_i - R, \tag{7}$$

21 where $R = \frac{(V_j(T) - V_j(1))}{V_{th}T}$. As $V_j(T) \geq 0$ and $V_j(1) = 0$, the remainder is non-negative $R \geq 0$. Since
22 $\sum_{i \in \mathcal{N}_j} w_{ij}^s r_i = \frac{C}{V_{th}T} > 0$, to mimic the relu function with positive input, we hope the remainder
23 $R$ is as small as posible. Thus, the ideal output spike count under $V_a > 0$ can be formulated as an
24 simple conditioned optimization problem:

$$\sum_{t=1}^{T} S_j^*(t) = \arg\min_x R$$

$$= \arg\min_x \frac{\sum_{i \in \mathcal{N}_j} w_{ij}^s \sum_{t=1}^{T} S_i(t) - x}{T} \qquad (8)$$

$$= \arg\min_x (\frac{V_a}{V_{th}} - x)$$

$$\text{s.t. } x \in \mathbb{N} \text{ and } x < \frac{V_a}{V_{th}}$$

25 Thus, for $V_a > 0$, the ideal spike count is

$$\sum_{t=1}^{T} S_j^*(t) = \lfloor \frac{V_a}{V_{th}} \rfloor. \qquad (9)$$

26 For $V_a \leq 0$, ideally the memberain potential should intergrate $C \leq 0$ for each timestep. With
27 $V_j(t-1) \leq 0$, as $V_j'(t) = V_j(t-1) + C \leq 0$, according to Eqn. 5, the ideal spike count should be
28 zero.

$$\sum_{t=1}^{T} S_j^*(t) = 0. \qquad (10)$$

29 Combine Eqn. 9 and Eqn. 10, the ideal spike count can be fomulated as follow:

$$\sum_{t=1}^{T} S_j^*(t) = \left\lfloor \max\left(\frac{V_a}{V_{th}}, 0\right) \right\rfloor. \qquad (11)$$

30 ### A.3  Proof of over-fire alleviation

31 In the buffered neuron, a new buffered potential $V^b$ is introduced:

$$V_j^b(t) = V_j^b(t-1) + V_{th} S_j(t). \qquad (12)$$

32 The spike generating process is described as below. Similar to the membrane potential, the initial
33 buffer potential $V_j^b(1)$ is set to zero.

$$S_j(t) = \begin{cases} 1, & \text{if } V_j'(t) > V_{th} \\ -1, & \text{if } V_j'(t) < 0 \text{ and } V_j^b(t-1) \geq V_{th} \\ 0, & \text{otherwise.} \end{cases} \qquad (13)$$

34 An over-fire situation can be described as the number of already generated spikes is greater than the
35 number of expected spikes. As the proposed buffered IF intervene the over-fire situation at spike
36 generation, we consider the over-fire situation at time $T_o$ before reset

$$\sum_{t=1}^{T_o} S_j(t) > \sum_{t=1}^{T_o} S_j^*(t) \qquad (14)$$

37 As $\sum_t S(t) \in \mathbb{N}$, we can rewrite Eqn. 14 as:

$$\sum_{t=1}^{T_o} S_j(t) \geq \sum_{t=1}^{T_o} S_j^*(t) + 1. \qquad (15)$$

The buffered potential $V_j^b(T)$ is derived by summing Eqn. 12 over time:

$$V_j^b(T_o) = V_{th} \sum^{T_o} S_j(t) + V_j^b(1) \geq V_{th}. \tag{16}$$

Since $T_o$ is reset before, the membrane potential at $T_o$ is

$$
\begin{aligned}
V_j'(T_o) &= V_{th} \sum_{i \in \mathcal{N}_j} w_{ij}^s \sum^{T_o} S_i(t) - V_{th} \sum^{T_o} S_j(t) \\
&= V_a(T_o) - V_{th} \sum^{T_o} S_j(t).
\end{aligned} \tag{17}
$$

For $V_a(T_o) > 0$, the definition of the floor function is

$$\frac{V_a(T_o)}{V_{th}} < \lfloor \frac{V_a(T_o)}{V_{th}} \rfloor + 1. \tag{18}$$

With $\sum^{T_o} S_j^*(t) = \lfloor \frac{V_a(T_o)}{V_{th}} \rfloor$ for $V_a(T_o) > 0$, we can infer the following according to Eqn. 15 and Eqn. 18,

$$
\begin{aligned}
V_j'(T_o) &= V_a(T_o) - V_{th} \sum^{T_o} S_j(t) \\
&= V_{th} \left( \frac{V_a(T_o)}{V_{th}} - \sum^{T_o} S_j(t) \right) \\
&\leq V_{th} \left( \frac{V_a(T_o)}{V_{th}} - \sum^{T_o} S_j^*(t) - 1 \right) \\
&< V_{th} \left( \lfloor \frac{V_a(T_o)}{V_{th}} \rfloor - \sum^{T_o} S_j^*(t) \right) \\
&< 0.
\end{aligned} \tag{19}
$$

For $V_a(T_o) \leq 0$, it is easy to tell that

$$
\begin{aligned}
V_j'(T_o) &= V_a(T_o) - V_{th} \sum^{T_o} S_j(t) \\
&\leq -V_{th} \sum^{T_o} S_j(t) \\
&< 0.
\end{aligned} \tag{20}
$$

Combining Eqn. 19 and Eqn. 20, we can conclude that $V_j'(t) < 0$ under over-fire situation. As shown above, constraints in Eqn. 13 are satisfied under an over-fire situation. A negative spike will be generated to reduce the total spikes count. Thus the over-fire problem can be alleviated.

## A.4 Strength-latency trade-off

The experiments on the CIFAR10 dataset (summarized in Table 1) exhibit the proposed strength-latency trade-off phenomena. Networks with the same representation power are converted from the same baseline ANN, which is trained under setting $S^{max} = 2$. It can be told that networks with the same representation power have similar performance while only half timesteps are required with $S^{max} = 2$. As the accuracy-latency trade-off still holds, the network with longer inference steps slightly outperforms the other one under the same representation power.

Table 1: CIFAR10 performance under same representation power with different settings. Top-1 accuracy is reported.

| $S^{max}$ | $T$ | ANN Top-1 accuracy | SNN Top-1 accuracy |
|---|---|---|---|
| 2 | 8 |  | 95.13% |
| 1 | 16 | 95.30% | 95.31% |
| 2 | 16 |  | 95.14% |
| 1 | 32 | 95.18% | 95.16% |

## A.5 Performance stability

To demonstrate the performance stability, series of experiments that share the same setting are conducted on both CIFAR10 and ImageNet datasets. Specifically, eight Resnet18 networks are trained and converted on the CIFAR10 dataset independently. Similarly, four VGG16 experiments are repeated on the ImageNet dataset. For concision, only Top-1 accuracy results are reported in Table. 2. Alone the average performance across multiple runs, standard deviation is also presented to illustrate the stability of the proposed method. Compared with results reported in the main paper, it can be anticipated that the the proposed method is universally reliable.

Table 2: Performance stability test classification tasks. Mean and standard deviation are reported as mean(std).

| Architecture | #Runs | ANN Top-1 Accuracy | SNN Top-1 Accuracy | $\Delta$ | $T$ |
|---|---|---|---|---|---|
| CIFAR10 | | | | | |
| ResNet18 | 8 | 95.47% (0.16%) | 95.19% (0.16%) | -0.15% (0.07%) | 8 |
| ImageNet | | | | | |
| VGG16 | 4 | 75.66% (0.15%) | 74.22% (0.19%) | -1.44% (0.09%) | 8 |

## B Object Detection With Spike Camera

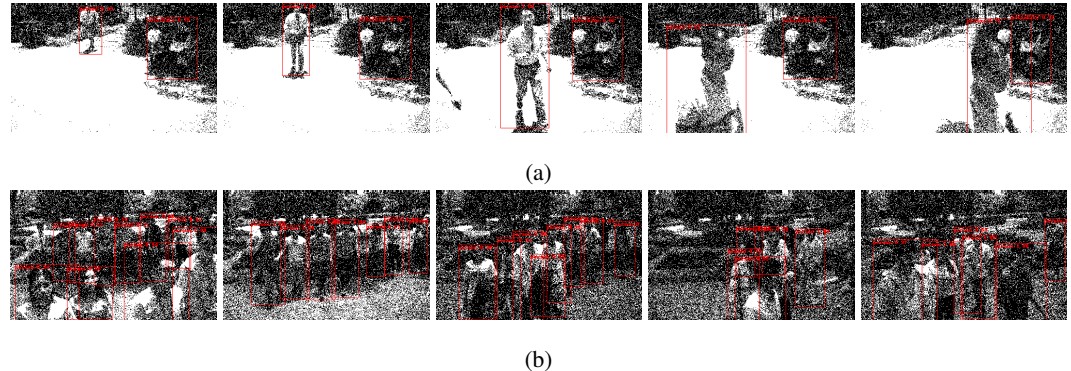

(a)

(b)

Figure 1: Object detection with spike camera. For each sequence, the binary input with the detection results is presented.

In contrast to conventional frame-based cameras which output the intensity of every pixel, event cameras represent the scene with events. As a typical type of event camera, Dynamic Vision Sensor (DVS) [6, 4] generates a spike asynchronously when there is a luminance change. Different from DVS cameras [6, 4], a Spike Camera [1] produces a spike whenever the accumulated intensity reaches a dispatch threshold. After that, the accumulator will be reset by subtracting the value of the threshold.

As the event generation schema in the spike camera closely resembles the neuron adopted in this literature, we directly apply the detection model trained with the VOC dataset on the steams collected by the spike camera. While Address Event Representation (AER) [9] is commonly adopted to deal with the asynchronously generated data, frame-like data is constructed from the spike stream which acts as the input to the SNN. The generated frame uses $\{0, 1\}$ to represents whether an event is observed in the pixel during a period.

Here we demonstrate two steams collected from a spike camera. As shown in Fig. 1, the converted SNN is able to detect objects with the spike camera even the original ANN is trained on conventional images.

## C   Detail experiment settings

As we only consider the ReLU activation, a hybrid inference framework is adopted to conduct experiments over different tasks. Most network inference is conducted using SNN, while the reset, such as the softmax layer in classification or post-processing layers in detection, is carried out by ANN. A readout layer is utilized to convert output spike trains to features. By scaling the parameters of linear layers properly, the expected real value feature simply becomes the spike rate of the output spike train $a_{out} = r = \frac{\sum^T S(t)}{T}$.

The identity connection in a residue block [3] prevents from directly applying weight normalization. To tackle this problem, we alter the conventional residual block by introduction a weighted add layer $WAdd(\cdot)$ where $WAdd(x, y) = W \odot (x + y)$. Here the $\odot$ denotes the element-wise multiplication. By setting the weight $W = 1$, the altered residual block function is identical to its conventional form.

Here we substitute all pooling layers with strided convolution layers. During inference, BatchNorm [5] acts as a straightforward linear layer that can be merged into a prior convolution layer. Thus BN layers are retained in the converted SNN.

For recognition tasks, both networks were trained with cosine learning rate scheduling whose initial learning rate is set to $\eta = 0.2$. Batch size was set to 256. Each model was trained for two stages: a fine-tuning stage after a training stage with 120 epochs for each stage. The fine-tuning stage uses the same setting as the training stage except the weights are inherited. For experiments on the CIFAR10 dataset, random erasing [10] was adopted other than standard augmentations. While for the ImageNet dataset, only standard data augmentation techniques were adopted. Label smooth [7] with $\epsilon = 0.1$ was utilized beside data augmentations. BatchNorm was added between convolution and activation layers for VGG16 architecture. For both models, Xavier initialization [2] was adopted. As QReLU was adopted, learning rate of its boundary parameter was set to $\eta^b = 0.02$ and $\eta^b = 0.002$ for ResNet18 and VGG16 respectively.

For detection tasks, the input dimension was set to $416 \times 416$, and batch size was set to 64 for both models. Multi-steps learning rate scheduler with an initial learning rate $\eta = 0.001$ was used during training. The model on the PASCAL VOC was trained with total $1 \times 10^5$ iterations where the learning rate decays by a factor $\beta = 0.1$ at $6 \times 10^4$ and $8 \times 10^4$ iterations. For the MS COCO dataset, models were trained using total $1 \times 10^6$ iterations where decays happen at iteration $8 \times 10^5$ and $9 \times 10^5$. Besides standard settings of Yolo, GIoU loss [8] was utilized for both models. Unlike classification tasks, the learning rate of boundary $\lambda^b$ was set the same as other parameters.