# OpenReview forum: "A High Performance and Low Latency Deep Spiking Neural Networks Conversion Framework"
_NeurIPS.cc/2022/Conference — NeurIPS 2022 Submitted_

### Official Review · Reviewer_UtSS · 2022-07-02

**Rating:** 3
**Confidence:** 4
**Soundness:** 2 fair
**Presentation:** 2 fair
**Contribution:** 1 poor

**Summary:**

This paper proposes to improve the ANN-to-SNN conversion method by several techniques including QReLU activation, buffered non-leaky-integrate-and-fire neuron, and multiple bits spike train. Experiments on both classification and detection tasks demonstrate strong performance.

**Questions:**

See Weakness above.

**Limitations:**

There is no potential negative societal impact.

**Strengths And Weaknesses:**

Strengths:
1. This paper not only considers the common classification task but also demonstrates strong results on the detection task.

Weakness:
1. The novelty of this paper is limited. The conversion error has already been theoretically analyzed in the literature [1,2,3,4]. Particularly, the inference error in this paper is studied as “unevenness error” in [3] and “deviation error” in [4] (while [4] is very recent and available after the submission deadline, [3] is earlier and should be noted). The quantized ReLU activation in this paper has been adopted in [3] as “quantization clip-floor activation”. The proposed buffered non-leaky IF neuron has been similarly adopted in the literature [5]. And the multiple bits spike train is inspired by existing literature except that this paper realizes it in a different way. So it looks like this paper simply combines multiple existing techniques and the novelty is very limited.

2. It is unclear how the proposed buffered neuron model and multiple bits spike train can be implemented on neuromorphic hardware. The compatibility with hardware should be discussed.

3. This paper does not use common neuron models and tries to enforce the inference of SNNs to be the same as quantized ANNs by proposing new neuron models. Then what is the advantage compared with directly applying well-established quantized ANNs that can also have hardware-efficient implementation? There should be some discussion and comparison.

[1] Deng, S., & Gu, S. (2020, September). Optimal Conversion of Conventional Artificial Neural Networks to Spiking Neural Networks. In International Conference on Learning Representations.

[2] Li, Y., Deng, S., Dong, X., Gong, R., & Gu, S. (2021, July). A free lunch from ANN: Towards efficient, accurate spiking neural networks calibration. In International Conference on Machine Learning (pp. 6316-6325). PMLR.

[3] Bu, T., Fang, W., Ding, J., Dai, P., Yu, Z., & Huang, T. (2021, September). Optimal ANN-SNN Conversion for High-accuracy and Ultra-low-latency Spiking Neural Networks. In International Conference on Learning Representations.

[4] Meng, Q., Yan, S., Xiao, M., Wang, Y., Lin, Z., & Luo, Z. Q. (2022). Training much deeper spiking neural networks with a small number of time-steps. Neural Networks.

[5] Thiele, J. C., Bichler, O., & Dupret, A. (2019, September). SpikeGrad: An ANN-equivalent Computation Model for Implementing Backpropagation with Spikes. In International Conference on Learning Representations.

---

> ### Author Response · Authors · 2022-08-02
> **Response to Reviewe UtSS**
>
> Thanks for the valuable information.
>
> **The novelty of this paper is limited.**
>
> We believe there is some misunderstanding about the novelty of our work.
> The main contribution of our proposed work is the theoretical analysis of the conversion error and its components and a novel high-performance conversion method to deal with the errors.
> Despite there exists several parallel works with a similar problem, there are significant differences between them.
> Here are the details.
>
> - **Compared with [1][2]**
> Differences between literature [1][2] and the proposed works have been discussed in the section introduction and related work. Quantitive performance between these works is also provided in the section experiment.
>
> - **Compared with [3]**
> We notice that the ''unevenness error'' in the parallel work[3] is indeed the same as the inference error in this work. However, the ``unevenness error'' is ignored during the analysis of conversion error (Eqn.11) in [3]. Different from these works, with an in-depth analysis of inference error, two subtypes are recognized in this literature and a novel neuron model is proposed to compensate for the overlooked error.
> As a result, our proposed method achieves a much better performance via dealing with inference errors.
>
> - **Compared with [4]**
> Later work in [4] proposed a method to compensate for the inference error via balancing the threshold in the neuron which is significantly different from the proposed method. Comparing the Imagenet results, it is interesting that the proposed method still enjoys a better performance.
>
> - **Compared with [5]**
> A similar but different inner state is used in [5]. However, in our work, the buffer is used to compensate for the inference error while in [5] the inner state is used to compute gradients.
>
> We did not claim to invent the multi-bit spike train and related works are properly cited. By integrating the multi-bit spike train, we propose a more general high-performance low latency conversion framework.
> In this general framework, the relationship between timesteps and maximal spike value named \textit{strength-latency tradeoff} is revealed and further confirmed by a quick experiment in the appendix.
>
> We argue that our method is of novelty and the proposed high-performance conversion method is of significance for ANN-SNN conversion approaches.
>
> [1] Deng, S., & Gu, S. (2020, September). Optimal Conversion of Conventional Artificial Neural Networks to Spiking Neural Networks. In International Conference on Learning Representations.
>
> [2] Li, Y., Deng, et al. (2021, July). A free lunch from ANN: Towards efficient, accurate spiking neural networks calibration. In International Conference on Machine Learning. PMLR.
>
> [3] Bu, T., et al. (2021, September). Optimal ANN-SNN Conversion for High-accuracy and Ultra-low-latency Spiking Neural Networks. In International Conference on Learning Representations.
>
> [4] Meng, Q., et al. (2022). Training much deeper spiking neural networks with a small number of time-steps. Neural Networks.
>
> [5] Thiele, J. C., et al. (2019, September). SpikeGrad: An ANN-equivalent Computation Model for Implementing Backpropagation with Spikes. In International Conference on Learning Representations.
>
>
> **It is unclear how the proposed buffered neuron model and multiple bits spike train can be implemented on neuromorphic hardware.**
>
> The multi-bits spike train has been hardware implemented in [45] cited in the literature. Since non-binary spikes are viable by hardware, the negative spike which can be viewed as a special case of a non-binary spike can be supported by hardware. Other than negative spikes, the proposed buffered neuron uses the same intrinsic operations such as integration and thresholding function as the basic IF neuron.
> Furthermore, we have discussed with some experts of Intel's Loihi and the micro-code engine in its architecture could support our algorithm.
> Thus, we believe the proposed buffered neuron should not post difficulty in hardware implementation which can be researched in future work.

---

> > ### Author Response · Authors · 2022-08-02
> > **Response to Reviewe UtSS(2)**
> >
> > **what is the advantage compared with directly applying well-established quantized ANNs that can also have hardware-efficient implementation? There should be some discussion and comparison.**
> >
> > As several works such as [1,2,3] mentioned earlier also enforce the output of SNNs to be the same as quantized ANNs, the problem can be asked to all these works. Actually, we believe the question of the advantage of SNN over quantized ANN can be asked to most SNN applications. Though this is beyond the scope of this literature, we try to summarize possible advantages. For SNNs, most computation happens when spiking neurons integrate input into their own inner memories. As computation and memory are tightly coupled, SNNs can naturally benefit from in/near memory computation compared to QANN. Moreover, as integration only happens when there are input spikes or events, spiking neurons naturally work on demand and unsynchronized while quantized ANN requires all inputs available to perform inference. As the proposed neuron has the same characteristics such as tightly coupled computation and memory, ability to work on demand, and unsynchronized, we believe the proposed framework shares the same advantage over ANNs with other SNNs.

---

### Official Review · Reviewer_HvVX · 2022-07-04

**Rating:** 4
**Confidence:** 4
**Ethics Flag:** Yes
**Soundness:** 2 fair
**Presentation:** 1 poor
**Contribution:** 3 good

**Summary:**

The paper describes a method to convert ANN with quantized ReLu activation functions into a spiking network emitting non-binary spike trains. The theory extends previous work on rate-based conversion from relu ANN to spiking networks: the authors characterize conversion error into sub-types. Beside the coding error (quantization error) they categorize the conversion error as under- and over-spike error and suggest methods to reduce them. One idea is to send signed spikes to signal when the accumulated inputs cross the threshold downward (effectively cancelling the previously transmitted spike and solving the over-spike error). The second is to transmit non-binary spikes with a spike amplitude quantized on a scale of powers of 2. It yields non-binary spiking networks with high-performance (for example 75% top1 accuracy on ImageNet  which is on par with current popular VGG or ResNet baselines).

**Questions:**

I wrote suggestions and questions in the Strengths and weaknesses section.

## Ethical concern:
The implementation of Yolo with DVS cameras is likely to converge towards low-consumption surveillance applications which raises ethical concerns. Although there are many applications of Yolo with DVS cameras which would not necessary lead there, the current Figure 4 demonstrates how to separate people from objects so it depicts quite explicitly how this technology can be applied for surveillance without mentioning the ethical concerns. There are many ways to reduce this negative impact, here are some concrete options: choose a different example in Figure 4 (and Figures from Section B in appendix) and publish the code so that independent researchers can understand better how to avoid misuses of the technology.

**Ethics Review Area:**

["Inappropriate Potential Applications & Impact  (e.g., human rights concerns)"]

**Limitations:**

I described limitations in the weaknesses section.

**Strengths And Weaknesses:**

## Strengths:
(1) The characterization of over- and under-fire conversion error is relevant and useful to design better conversion schemes of spiking networks to non-spiking networks.

(2) The performance is very high and it bridges the gap between spiking networks and regular quantized neural networks.

(3) The reduction of  over-fire error with negative spikes seems to have a large positive-effect on the accuracy and its description is understandable.

## Weaknesses:
(1) The paper is poorly written: most importantly the main text lacks of mathematical rigor and uses confusing notations. For instance in section 3.1, it is not clear what is a definition of a notation for $S^*(t)$. The closest to a definition would be equation (4) but this defines the sum over $t$ and not $S^*(t)$ per-se which is confusing. So the concepts of "ideal conversion" is defined in a cumbersome and complicated way although it seems quite important. Also the definitions of $r_i$,  $S^*(t)$ and $r^*_i$ are unclear and maybe redundant. Because of this I did not understand why a proof mentioned at line 135 is necessary, I read through the proof in appendix anyway and it seems to rely on some wrong arguments: for instance it is written in the appendix "For $V_a$ > 0 we infer $C$ > 0" but C depends on $t$ although $V_a$ is a quantity integrated overtime so $V_a > 0$ cannot imply $C > 0$. This statement is either wrong or unclear, although it is probably a minor argument for the proof, it shows the general lack of rigor in the theoretical section. I think this paper will be much better if the theory section is re-written, mostly sections 3.1, 3.4 and this proof which are hard to read and unclear at the moment.

(2) The implementation is not clear and the code is not provided, this is a real problem because details are missing to fully understand the algorithm. Concretely I see two possible ways of implementing this model but I do not know which one is implemented. It is not the same if the layer 2 fires spikes S(t) right after layer 1 S(t) or whether layer 2 waits the end of the computation S(T) of layer 1 before computing S(1).
- First possibility: the code can be something rather simple like replacing the ReLU or QReLU with a spiking version of it which receives the integrated sums of spikes and outputs the integrated sum of spikes, if this is what is implemented it should be acknowledged clearly and the the theoretical section should clearly acknowledge that the latency accumulates at each layer. Also, if this is the implementation the activity is buffered at each layer so the necessity of the additional spike buffer is probably unnecessary.

- Second possibility: the model is implemented as a sort of recurrent neural network where the spikes S(t) at all layers are computed before sending them to S(t+1). However, training a recurrent network in this way would require to use back propagation through time which typically multiply the memory consumption of the algorithm by a factor T and raises a clear drawback to train large ImageNet or VOC models on GPUs. Considering the trade off that is therefore necessary between T and batch size, it should at least be debated somewhere.

In any case, even if the implementation is done in another way which I did not describe here. This should be acknowledged and described clearly. Ideally the code should be shared so that any ambiguity can be lifted by reading the code directly.

(3) I think that the theoretical statement that the negative spike can correct the over-fire problem is only true with a single hidden layer. A clear counter example with two layers can be designed where a neuron in hidden layer 1 spikes positively at time step 1 (S(1)=1) which engages non linear computations in the hidden layer 2 at the same time step, and then spikes negatively at time step 2 (S(2)=-1) leading to a QReLU output of 0. Assuming the second possible implementation that I described in the paragraph above it is not guaranteed that the accumulated effects of the negative and positive spikes cancel out because of the non-linearity in layer 2. Unfortunately I cannot be sure whether this can occur in the simulation because of the missing information about the implementation. If it does occur, the imperfections of the negative spike solution of over-fire errors should be acknowledged in the main text.

(4) By introducing the concept of non-binary spike trains the conceptual boundary between spiking neurons and quantized ANN is blurred. This should be acknowledged clearly and the remaining difference and benefits of non-binary SNN and Q-ANN should be discussed. About their similarities, one could acknowledge:
- This conversion method requires a pre-training with a well chosen QReLU activation function so it requires the similar training algorithm as a quantized ANN (say a binary feed-forward neural network trained with straight through estimators).
- Certainly most current hardware capable of simulating a non-binary SNN could also implement a Q-ANN. Which raises questions about the benefits of the spiking solution.

About there differences, I would love to see some quantitative comparisons, for instance:
- Can one add the Q-ANN into the energy ratio plot ? (not sure how to count integer multiplications into the plot relying on the number of additions, but this was done somehow to integrate spikes with amplitude larger than one in this plot already so why not?). If not, finding common metric to estimate to energy consumption of Q-ANN and non-spiking network would be great in this context but it might be left for future work.

(5) Details of implementation about the Yolo architecture are not provided. It probably requires to implement non-binary communications on some unusual blocks like residual blocks. Although a paper is cited for this purpose it is necessary to explain whether something similar or different is used. I also think that this can be problematic in regards to the theory section depending on the implementation, so a clear description of the implementation of residual block using the notations and concepts from the theory section would be strongly welcome. Again, one option is to publish the code.

---

> ### Author Response · Authors · 2022-08-02
> **Response to Reviewer HvVX**
>
>
> Thanks for the detailed review and helpful suggestions.
>
> **Notation problem of the manuscript**
>
> $S^*(t)$ is the output spike train under the assumption that the neuron integrates membrane potential uniformly, which can be viewed as a special case of predefined $S(t)$.
> Since rate base coding is adopted, only the integration of time matters thus Eqn.4 is given to analyze the coding error.
> To make it more clear, we will rewritten line 133 as ``For an ideal conversion, the spiking neuron integrates membrane potential uniformly across time and introduces no error during inference. Under this ideal condition, the output spike count should fulfill the following equation. ''. Similar, the title of section 1.2 in the appendix will be revised to "proof of output spike count under ideal condition"
>
> $r_j$ is the rate of a spike train which is defined in line 124. Thus $r^*_j$ is the rate of the spike train under the assumption that the neuron integrates membrane potential uniformly.
>
> As the proof in the appendix is given under the assumption that membrane potential is accumulated uniformly (line 16 in the appendix), it is easy to infer that $C>0$ under the condition that $\sum_t{C}=V_a>0$
>
> We will improve the presentation of the manuscript accordingly.
>
> **Lack of implementation**
>
> We will release the source code once the paper gets published.
> Please note that this literature describes a conversion method which means the learning (backpropagation) only happens in ann regime.
> Since no backpropagation in SNN is required, drawbacks of RNN training do not apply to this work.
> For simulation of SNN, both the implementation methods described are viable.
> However, the proposed neuron does not require the whole input spike train to compute output, thus there should be no latency accumulation problem.
>
> **Limitation of negative spikes**
>
> The proposed buffered IF neuron aimed at resolving the over-spike problem for an arbitrary spiking neuron via compensating the output spike train to match the spike count under ideal conditions (Eqn.4). As no assumption is made for the input nor the output neuron, the proposed neuron works universally across the network.
> Moreover, please note that QReLU is adopted in ann training, while buffered IF neurons are used in the SNN inference.
>
> **non-binary spike trains blur the boundary of quantized ANN and SNN**
>
> Actually, as described in the section coding error and section quantized ReLU, the equivalence between quantized ANN and SNN is illustrated.
> Also, the coding error is completely eliminated by mapping weights from quantized ReLU-enabled ANN. The backpropagation methods are given in the PACT algorithm which can be viewed as an enhanced STE algorithm.
>
> To enjoy the metric of SNN, the inference should happen on neuromorphic hardware. We believe it is fairer to compare ANN and SNN energy consumption or inference latency on quantized ANN optimized hardware and neuromorphic hardware. Since we are unable to access neuromorphic hardware, a simple yet representative efficiency analysis is provided. Please note that this count-based efficiency analysis is heavily in favor of ANN as expensive memory access is ignored.
>
> **Details of implementation about the Yolo architecture are not provided**
>
> Detail experiment setting can be found in the appendix section. To deal with the identity connection in the residual block, the weighted sum (line 84 in the appendix) is used to replace the conventional sum operating. Please note that summation in the residual block is indeed a linear operation that does not affect the theoretical analysis of the neuron.
>
> **Ethical concern**
>
> Thanks for the suggestion. As the demo is essential to a spiking version of YOLO, it should be viewed as a common object detector. We will remove these results from the main manuscript.

---

> > ### Comment · Reviewer_HvVX · 2022-08-07
> > **Conclusion**
> >
> > Thank you for the clarification.
> >
> > I will increase my grade from 3 to 4, because I think the authors have understood a non-trivial problem and described a method that makes sense and remains novel to me (even-though the other reviewers have reported missing recent citations). Improving the clarity of the technical description of the method would be the most important to make a great publication in my opinion.
> >
> > There remain also obscure concepts in the math and the algorithms that I do not fully grasp. Despite the authors comments I still do not understand how the skip-connection in the ResNet operation can be implemented. If those neurons are different unit in the hardware one cannot simply send the voltage information since the information has to be transmitted with spikes.
> >
> > Sorry but I did not request to remove the Yolo results, acknowledging the ethical concern and it is now common in the conference and publishing the code would have been sufficient in my opinion.

---

> > > ### Author Response · Authors · 2022-08-09
> > > **Thanks very much for the comment**
> > >
> > > Thanks for your valuable reply. We are very grateful and appreciate your recognition of the novelty and the significance of our proposed work. To help readers understand our work and contribute to the community,  we will definitely improve the clarity of the manuscript and release the source code once the paper gets published.
> > >
> > > For the skip-connection problem, our ResNet experiments are conducted on our own developed simulator. The proposed method has NOT been tested on hardware implementation yet. We are not sure whether there exists any hardware capable of running such a big network, e.g. ResNet. We are trying our best to cooperate with hardware experts and validate our proposed work on real chips in the future.
> > > In terms of the theoretical part, skip-connection is also a subset of the fully connected network. As our proposed work focuses on neurons, the connection including the skip connection is hardly related to the contribution of this work.
> > >
> > > Thanks for pointing out the ethical concerns. We would publish our source code once the paper gets publishing.

---

### Official Review · Reviewer_PgR1 · 2022-07-11

**Rating:** 2
**Confidence:** 5
**Soundness:** 1 poor
**Presentation:** 3 good
**Contribution:** 1 poor

**Summary:**

The authors present a well-designed ANN-to-SNN conversion method. They claimed to discover and discuss the overlooked conversion errors: coding error and inference error. A QReLU activation for the training phase and a buffered none-leaky neuron for the inference phase were proposed. The authors gave good results on classification and detection tasks. To some extent, the authors did not conduct detailed literature research and over-claimed their contributions.

**Questions:**

1.The idea of QReLU is similar to paper [1] & [2]. Please demonstrate the differences between them. Comparisons of the performance should also be included.
2.V_(T_o) < 0 is equivalent to the over-fired case. My question is: Does this condition restrict that there is only one negative pulse before T_o? And, given that the final membrane potential may not exceed Vth, how can the effects of the over-fired situation be mitigated?
3.I don't understand how the experiment of spike camera detection is implemented. Please specify more details, including how long the sequence you input to the SNN and how the data used was obtained.
4. What does the time step of ANN in Fig.2 mean?
5. Line 205 says that the computational-expensive multiplication operation needs to be avoided. Please explain how log2 operation in the newly designed neuron will be implemented in hardware?
6. Line 197 says "we integrate shift-based multi-bits spike", I didn't find a statement about "shift" in the paper. What is "shift"?
7. There is no ablation study on “QReLU” and “Multiple Bits Spike Train” found in the main paper.
8. If λ is learnable, does T need to be pre-set in QReLU? Can the performance of different time steps be provided under the same model?
9. As the aim of this work is to get loss-free conversion, if the buffered potentials are the integration of all inputs across time, why not just output the buffered potential to get loss-free conversion.

**Limitations:**

This paper does not do enough quantitative analysis and explanation of the proposed negative spike method. Neurons that generate negative spikes belong to the special category of spiking neurons.
Since the authors use a new type of neuron, and the new neuron is obviously more complex than the traditional neuron, the authors should include more analysis about the power consumption and hardware implementation of this neuron.

**Strengths And Weaknesses:**

Strengths:
1. This paper is easy to follow. The authors introduce the QReLU function, buffered none-leaky neurons to promote accuracy and integrate the shift-based multi-bits spike train to increase representation efficiency.
Weakness:
1. The major weakness is the novelty. The main contributions are the error identification and a quantization ReLU activation, both of which have been proposed in some previous work [1], [2]. Also, the negative spike has been proposed by spiking-yolo. Please clarify the difference, or is this work a combination of these techniques?
2. Experiments are not enough to demonstrate the effectiveness of the proposed method. For example, it is mentioned in line 177 that λ is learnable, according to Eqn. 9. Please clarify whether T needs to be set. If T needs to be set, what is it equal to?
3. The authors use a newly proposed neuron model which may not be supported by the current hardware, which will diminish the significance of this paper.
4. There are some ambiguous expressions in the paper that are hard to understand.

[1] Yan, Z., Zhou, J., & Wong, W. F. (2021). Near lossless transfer learning for spiking neural networks. In Proceedings of the AAAI Conference on Artificial Intelligence.
[2] Bu, T., Fang, W., Ding, J., Dai, P., Yu, Z., & Huang, T. (2022). Optimal ANN-SNN Conversion for High-accuracy and Ultra-low-latency Spiking Neural Networks. In International Conference on Learning Representations.

---

> ### Author Response · Authors · 2022-08-02
> **Response to Reviewer PgR1**
>
> Thanks for providing the detailed review and valuable information.
>
> **Lack of novelty. Similar to [1][2][3]**
>
> We believe there are some misunderstandings about the novelty of our work.
> The main contribution of our proposed work is the theoretical analysis of the conversion error and its components and a novel high-performance conversion method to deal with the errors.
> Despite there exists several parallel works with a similar problem, there are significant differences between them.
> Here are the details.
>
> - **Compared with [1].**
> In [1], the authors enforce a [0,1] feature value during ANN training while we let the boundary be learned in the proposed QReLU.
> This literature reveals a general equivalency between quantized ANN and SNN, which enables the conversion approaches continuously benefits from future advances in quantized-aware training techniques.
> Moreover, the inference error is overlooked in [1]. Please note that [1] is already compared in the section experiment.
>
> - **Compared with [2]**
> We notice that the ''unevenness error'' is indeed the same as the inference error in this work.
> However, the ``unevenness error'' is ignored during the analysis of conversion error in [2].
> As a result, our proposed method achieves a much better performance via dealing with inference errors.
>
> - **Compared with [3]**
> Different from spiking yolo[3] which uses negative spikes to resemble the leaky ReLU in ANN, our proposed method utilizes the negative spike to compensate for inference error. The focus is still on the conversion error, not the leaky ReLU.
>
>
> We argue that our method is of novelty and the proposed high-performance conversion method is of significance for ANN-SNN conversion approaches.
>
>
> [1] Yan, Z., Zhou, J., & Wong, W. F. (2021). Near lossless transfer learning for spiking neural networks. In Proceedings of the AAAI Conference on Artificial Intelligence.
>
> [2] Bu, T., Fang, W., Ding, J., Dai, P., Yu, Z., & Huang, T. (2022). Optimal ANN-SNN Conversion for High-accuracy and Ultra-low-latency Spiking Neural Networks. In International Conference on Learning Representations.
>
> [3] Kim, S., Park, S., Na, B., & Yoon, S. (2020). Spiking-YOLO: Spiking Neural Network for Energy-Efficient Object Detection. Proceedings of the AAAI Conference on Artificial Intelligence, 34(07), 11270-11277.
>
> **it is mentioned in line 177 that λ is learnable, according to Eqn. 9. Please clarify whether T needs to be set. If T needs to be set, what is it equal to?**
>
> As demonstrated by Algorithm 1 in the appendix, $T$ is a hyperparameter. This hyperparameter is used both in QRelu during ANN training and SNN inference. That's also the reason why ANN has a hyperparameter $T$ in Fig.2.
>
> **The authors use a newly proposed neuron model which may not be supported by the current hardware, which will diminish the significance of this paper.**
>
> The proposed neuron shares the same intrinsic operations such as integration and thresholding with the original IF neuron.
> As far as we know, intrinsic operations such as multiple states and shift operations are supported by the micro-code engine of Loihi.
> We would further extend our proposed method in the hardware to test its capability in the future.
>
> **$V{(T_o)} < 0$ is equivalent to the over-fired case. My question is: Does this condition restrict that there is only one negative pulse before $T_o$? And, given that the final membrane potential may not exceed Vth, how can the effects of the over-fired situation be mitigated?**
>
> Actually, $T_o$ is assumed the moment the over-fired situation happened, where $V'_j(t) < 0$ is the necessary but not sufficient conditions of the over-fired situation. As the over-fired case can happen in conventional IF neurons, negative spikes are not necessary.
>
> Intuitively, the over-fired situation described a phenomenon when a neuron fires more spikes than expected. In the over-fired situation, the membrane potential is lower than expected (actually lower than zero as shown) as more spikes are fired. The proposed neuron fires negative spikes to compensate for previous positive spikes and increases the membrane potential to alleviate the problem.
>
>
> **What does the time step of ANN in Fig.2 mean?**
>
> $T$ is a hyperparameter used in QReLU.
>
> **Line 205 says that the computational-expensive multiplication operation needs to be avoided. Please explain how log2 operation in the newly designed neuron will be implemented in hardware? Line 197 says "we integrate shift-based multi-bits spike", I didn't find a statement about "shift" in the paper. What is "shift"?**
>
> Log2 operation can be simply replaced with conditional functions under bounded $S^{max}$, we use log2 just to keep the representation clean. The shift is the bitwise operator. To make it clear, we will add a description "The adjust function can be implemented with a conditional function." in line 206.

---

> > ### Author Response · Authors · 2022-08-02
> > **Response to Reviewer PgR1(2)**
> >
> > **There is no ablation study on “QReLU” and “Multiple Bits Spike Train” found in the main paper.**
> >
> > As mentioned in the caption of Fig.3 section ablation study, Q-ANN denotes ANNs trained with QReLU. Ablation study of multi-bit spike train can be found in the appendix section strength-latency trade-off.
> >
> > **If λ is learnable, does T need to be pre-set in QReLU? Can the performance of different time steps be provided under the same model?**
> >
> > $T$ is a hyperparameter. An ablation study on $T$ can be found in the section ablation study.
> >
> > **As the aim of this work is to get a loss-free conversion, if the buffered potentials are the integration of all inputs across time, why not just output the buffered potential to get a loss-free conversion.**
> >
> > Actually buffered potential is the integration of outputs. However, assuming potential can be output, we believe output accumulated membrane potential $V_a$ is more likely to get a loss-free conversion.
> >
> > If the potential can be propagated, the SNN just becomes a conventional ANN which diminishes the advantage of SNN such as on-demand computation, and in-memory computation.

---

> > ### Comment · Reviewer_PgR1 · 2022-08-09
> > **Thanks very much for your response**
> >
> > However, the responses do not address my concerns. I still believe the major weakness is the novelty. The main contributions have been proposed in some previous work. Thus, I decide to keep my score.

---

> > > ### Author Response · Authors · 2022-08-10
> > > **Thanks for your reply**
> > >
> > > We are very grateful for your straightforward reply. But, as we have addressed all differences between the proposed work and all papers mentioned in the review comments, could you be a little bit more specific about the novelty problem?
> > >
> > > We understand that the novelty of the proposed work degrades from a novel problem with a novel solution to a novel solution to a known problem, but, we still argue that our novelty would contribute to the SNN community definitely. We have revised the manuscript accordingly to remove all statements of a novel problem and updated it in the OpenReview system.
> > >
> > > Your advice and suggestion will be very helpful to our further improvement. Many thanks in advance!

---

> > > > ### Comment · Reviewer_PgR1 · 2022-08-10
> > > > **The novelty problem**
> > > >
> > > > The authors claim three major contributions in their paper. The first is the analysis of conversion errors, which actually has been proposed in [1,2]. The second is the methods to eliminate two types of conversions. However, the idea of using quantized ReLU activation to eliminate coding error has been proposed in [1]. The new work is to use the negative spike that has been proposed by [3] to eliminate inference error. Thus, I believe the novelty of this paper is limited, which is far from the standard of NeurIPS.
> > > >
> > > >
> > > > [1] Bu, T., Fang, W., Ding, J., Dai, P., Yu, Z., & Huang, T. (2022). Optimal ANN-SNN Conversion for High-accuracy and Ultra-low-latency Spiking Neural Networks. In International Conference on Learning Representations.
> > > >
> > > > [2] Meng, Q., Yan, S., Xiao, M., Wang, Y., Lin, Z., & Luo, Z. Q. (2022). Training much deeper spiking neural networks with a small number of time-steps. Neural Networks.
> > > >
> > > > [3] Kim, S., Park, S., Na, B., & Yoon, S. (2020). Spiking-YOLO: Spiking Neural Network for Energy-Efficient Object Detection. Proceedings of the AAAI Conference on Artificial Intelligence, 34(07), 11270-11277.

---

### Review · Ethics_Reviewer_RTtR · 2022-08-01

**Recommendation:**

The paper should acknowledge the possibility of ANN-to-SNN conversion furthering the application of video surveillance. I would focus on the salient characteristics of SNNs and the spike cameras/dynamic vision sensors (DVS) that they are especially compatible with. For example, their lower energy usage may make them deployable in more challenging environments or in greater numbers. Do they also have greater low-light capability?

The paper should also describe how the spike camera images in Figure 4 were obtained. If they were obtained from another party, that party should be acknowledged. Regardless of the source, more information about the scene such as time, place, and other circumstances would be good. The issues noted above, of whether the people shown were aware of and consented to the recording, should also be discussed, whether the answers are positive or negative. Even though the images are grainy, perhaps it would be better to blur faces regardless.

Reviewer HvVX raised the issue of surveillance and also suggested two mitigations: choosing different example images and publishing the code. I agree with the second but not the first: choosing a different example does not make the issue go away and may conceal it. A different example may be good to show readers another potential application, but I would show it in addition to the current figures, not in place of.

**Ethical Issues:**

Yes

**Ethics Review:**

This paper is on converting artificial neural networks (ANNs) to spiking neural networks (SNNs), motivated by increased energy efficiency. One of the applications presented in the paper is visual object detection, and specifically, detection of humans in video collected by a spike camera. Although the video frames shown in Figure 4 (and Figure 1 in Appendix B) are grainy, they are suggestive of street scenes as might be captured by a surveillance camera. This raises the issue of whether the paper's advance could further enable surveillance, for both good and bad. It also raises the issue of how the images in Figure 4 were collected, whether the people shown in them were aware of the collection, and whether they consented to the recording.

---

### Review · Ethics_Reviewer_GUDG · 2022-08-11

**Recommendation:**

The authors have removed results that earlier reviewers pointed out that would specifically point to this. In addition to this, it would be good to acknowledge and discuss the potential downsides of such applications and cite research demonstrating the sensitivity of such applications.

**Ethical Issues:**

Yes

**Ethics Review:**

This paper proposes methods that make it more efficient and fast to run computer vision models- including tasks such as object detection models, classification tasks etc.

This can make the adoption and upkeep of CV models for various tasks such as surveillance more attractive as well. The authors have removed results that earlier reviewers pointed out that would specifically point to this. In addition to this, it would be good to acknowledge and discuss the potential downsides of such applications and cite research demonstrating the sensitivity of such applications.

---

### Author Response · Authors · 2022-08-02
**Overall Responses to Controversial Comments**

We thank all reviewers for their time and valuable comments. The main controversial points of reviewers' comments can be summarized as follows,
- Lack of novelty since several published papers have some similar perspectives compared with our proposed work.
- Hardware compatibility issue as our proposed method uses a two-state-buffer model for neurons compared with the conventional single-state-buffer one.

After reading all comments very carefully, we believe there are some misunderstandings during the review process. Indeed, we try to solve a very similar problem for the ANN-SNN conversion. But, our proposed method is very different from those papers mentioned in the comments. In fact, our first version of the manuscript has been finished and submitted in Jan. 2021. Unfortunately, previous reviewers were not appreciating such a problem. We are glad that several papers have been published to solve the same problem, which indicates the importance and significance of conversion errors. Our proposed method still has a state-of-the-art performance compared with these works and we argue that our approach has a better solution and insight to the problem. We give the detailed difference between our proposed work and others in the following response comments.

For the hardware concerns, we have discussed with some experts of Intel's Loihi and the micro-code engine in its architecture could support our algorithm. We would further extend our proposed method in the hardware to test its capability in the future. We hope that our replies can be considered and helpful in the final evaluation of this paper. Both manuscripts have been revised accordingly.

---

### Meta-Review · Area_Chair_r7rr · 2022-08-23

**Recommendation:** Reject
**Confidence:** Certain

**Metareview:**

Spiking Neural Networks (SNNs) have some advantages, especially in terms of power consumption, over standard Artificial Neural Networks (ANNs). However, most trained networks are ANN and therefore this work presents a conversion scheme from SNNs to ANNs with some desired properties. Unfortunately, the reviewers found the contribution of this work insufficient in terms of novelty. The authors argued, in the rebuttal, that there are differences between prior art and the current paper. The reviewers acknowledged these differences but were not convinced that the differences are significant enough. The authors also noted that this work was sent to publication in Jan 2021 and was rejected while some of the relevant papers were published after this date. We understand the frustration that this situation is likely to generate. However, when reviewing this work the relevant date is the deadline for NeurIPS submission deadline and therefore these studies should be compared to in the paper.

**Award:**

No

---

### Decision · Program_Chairs · 2022-09-14

Reject